# Electrical Properties of Textiles Treated with Graphene Oxide Suspension

**DOI:** 10.3390/ma14081999

**Published:** 2021-04-16

**Authors:** Danil Valeriyevich Nikolaev, Zakhar Ivanovich Evseev, Svetlana Afanasyevna Smagulova, Irina Veniaminovna Antonova

**Affiliations:** 1“Graphene Nanotechnology” Laboratory, Physical-Technical Institute, North-Eastern Federal University in Yakutsk, 677000 Yakutsk, Russia; dv.nikolaev@s-vfu.ru (D.V.N.); sa.smagulova@s-vfu.ru (S.A.S.); 2Laboratory of Physics and Technology of Three-Dimensional Nanostructures, Rzhanov Institute of Semiconductor Physics SB RAS, 630090 Novosibirsk, Russia

**Keywords:** graphene, graphene oxide, e-textiles, wearable electronics

## Abstract

Two-dimensional nanomaterials such as graphene can provide various functional properties to textiles, which have great potential in sportswear, healthcare etc. In this study, the properties of nylon and cotton-based electronic textiles coated with reduced graphene oxide are investigated. After reduction of graphene oxide coating in hydrazine vapor, e-textiles with a resistance of ~350 Ω/sq for nylon, and ~1 kΩ/sq for cotton were obtained. Cyclic mechanical bending tests of samples showed that the resistance increases during bending up to 10–20%. The use of bovine serum albumin as an adhesive layer improved the wash stability for samples with nylon up to 40 washing cycles. The use of BF-6 glue as a protective layer reduced changes in resistance during bending, and improved wash stability of cotton samples. It was shown that the resistance of the obtained samples is sensitive to changes in temperature and humidity. In addition, obtained e-textiles attached to a person’s wrist were able to measure heart rate. Thus, the obtained electronic textiles based on cotton and nylon coated with reduced graphene oxide demonstrates good characteristics for use as sensors for monitoring vital signs.

## 1. Introduction

Presently, thin-film materials for wearable electronics have a wide potential for application as embedded sensors of vital signs, flexible wearable batteries, heating elements [1,2,3], etc., which attracts a high interest of the scientific community on this topic. The development of technologies for the production of new two-dimensional materials and their application for conductive textiles are important scientific and practical tasks [4]. In particular, there are many research articles on the use of carbon nanotubes (CNTs) [5], conducting polymers [6], metal nanofibers [7] and nanoparticles [8], graphene, and its derivatives [9]. They can be used for functionalization of both natural and artificial fibers. However, the safety and biocompatibility of carbon nanotubes are under discussion [10]. It is considered that the sharp edges of CNTs can damage cell membranes and DNA, which limits their use as modifiers for wearable e-textiles. Metal nanofibers and nanoparticles also raise certain concerns about biocompatibility and environmental safety [11], and the functionalization of textiles with metal nanoparticles does not provide sufficient conductivity for practical use in wearable electronics [8]. Electrically conductive polymers are expensive and difficult to apply to organic textile materials [6]. E-textiles based on graphene and its derivatives, such as graphene oxide (GO), has been a focus of much attention due to their high conductivity, resistance to mechanical stress, and biocompatibility [12,13,14]. In addition, GO can form stable aqueous suspensions, due to oxygen functional groups on the surface of GO [15]. This simplifies many technological processes associated with the deposition of GO. Also, oxygen groups bind GO flakes via Van-der-Waals forces with many natural and artificial fibers, thereby increasing their stability to washing and mechanical stress [16,17,18].

The reduction method is an important factor in e-textiles based on GO. Chemical reduction is widely used for GO e-textiles. For example, in the study [19] GO on acrylic threads was reduced by immersion in an aqueous solution of 0.5 wt.% sodium hydrosulfite for 30 min. For the obtained textiles, the electrical resistance was about 10^2^ Ω/cm. In the study [20] GO were applied to a nylon/polyester fabric and reduced in sodiumborohydride (NaBH4). This method made it possible to obtain resistances of the order of 100 kΩ/m^2^. Shateri-Khalilabad et al. [21] has compared various reducing agents and showed that the use of Na_2_S_2_O_4_ and N_2_H_4_ yielded the best results in conductivity and tensile strength for cotton/GO e-textiles, with the obtained surface resistances of 19.4 and 62.7 kΩ/cm, respectively. In study [22], the GO on cotton was reduced in hydrazine hydrate vapors for 30 min. This approach made it possible to obtain a very low surface resistance, only 40 Ω/sq.

Application of nanomaterials can significantly improve tensile strength of the composite materials. For example, a study [23] reports that adding CNTs to rubber can increase the tensile strength by up to 105% depending on concentration. The simultaneous application of multiple nanomaterials can provide the benefits from both materials. V. Kumar [24] reported that the addition of nanographite and CNTs improved the properties of the piezo-resistive strain sensor and actuator based on silicone rubber compared to pure nanographite and CNT additives. In addition, in the study [25] it is reported that the addition of silver nanoparticles and GO improved the tensile strength of nylon fibers compared to nylon coated only with reduced graphene oxide (rGO). Bovine serum albumin (BSA) protein can be used as a universal adhesive an improve the adsorption of GO on the surface of nylon and cotton fibers. This method is now widely used with materials such as nylon, cotton, and polyester [17]. BSA is an amphiphilic protein that can connect to the different materials through hydrophobic (non-polar) and hydrophilic (polar) interactions [26,27].

Most studies have shown that the resistance of e-textiles based on GO increases significantly after several wash cycles [9,12,20,28]. Therefore, studies aimed at improving stability and practical applicability of e-textiles based on GO is important. In this study, we compared nylon and cotton e-textiles with a GO coating. The stability of such coatings to mechanical stress and washing has been demonstrated. The use of BSA as the adhesive layer improved the wash resistance of the nylon samples. In comparison, BSA did not significantly improve the adhesion of GO to cotton. For the first time, the effect of a protective coating based on BF-6 was investigated. It was found that the use of high concentrations of GO (3 mg/mL) allowed for the significant reduction of the change in electrical resistance after stretching the nylon fabric. It was demonstrated that the samples can be used as temperature, humidity, and heart rate sensors. The obtained e-textiles can be used as sensors for monitoring vital signs in the Arctic conditions.

## 2. Materials and Methods

Commercially available nylon and cotton have been used as a basis for conductive textiles. The nylon macromolecules [—HN(CH_2_)_6_NHOC(CH_2_)_4_CO—]*_n_* are generally electrostatically neutral, but have negative (oxygen atom of the carbonyl group) and positive (hydrogen atom of the amide group) charges. Cotton is 95% cellulose, the remaining 5% is fat and mineral impurities. Cellulose is a natural polymer, the monomers of which are glucose molecules. The structural formula of cellulose shows that each monomer includes three hydroxyl groups.

A GO suspension was prepared with the modified Hummer’s method described in a previous study [29]. The difference from the conventional Hummer’s method is that the method used in this study does not involve the use of an ice bath and ultrasonic decomposition of graphite oxide and intercalation is achieved by increasing the mixing time.

To obtain electrically conductive textiles, both cotton and nylon samples were covered with a GO layer by dipping into a GO aqueous suspension. The reduction of GO was performed in hydrazine vapor on a sealed water bath at a temperature of 60 °C, for one hour. Three types of samples were prepared and investigated: samples containing only rGO, samples with an intermediate layer of bovine serum albumin (BSA/rGO), samples with an additional layer of BF-6 glue (rGO/BF-6 and BSA/rGO/BF-6). In the second case, BSA (BSA/rGO) was used to improve the adhesion of GO to textiles. A BSA solution was obtained by adding 125 mg of dry BSA powder to 50 mL of water under the constant stirring at 80 °C. The resulting solution was acidified with hydrochloric acid to pH = 3. In the third case, a protective layer of BF-6 (rGO/BF-6 and BSA/rGO/BF-6) glue was applied to the rGO layer, which is used for gluing various flexible materials, for example textiles, leather, as well as in medicine for treating microtraumas. The glue includes phenol-formaldehyde resin, polyvinyl butyral, dibutyl phthalate, castor oil, rosin, dissolved in ethyl alcohol. The protective layer of BF-6 can protect the sample surface against damage caused by mechanical deformations, preventing premature peeling of rGO particles from fabric fibers. The number “6” gives the percentage of polyvinyl butyral in dry matter. The general view of the textile samples is shown in Figure 1.

The main stages of obtaining conductive textile samples included:Nylon and cotton samples with a size of 1 cm × 4.5 cm were dipped in an acidified BSA solution and dried at room temperature. 5 mL of BSA solution was used for 14 samples.The nylon and cotton samples with BSA layer were immersed into the 1 and 3 mg/mL GO suspensions under stirring and dried at room temperature. The GO concentration was determined by weighing the dry residue. 5 mL of the GO suspension was used for 14 samples.Obtained fabrics with dry GO layer were treated with hydrazine vapor on a water bath for one hour at a temperature of 60 °C in a sealed container.Some of the samples were covered with a thin layer of BF-6 glue from both sided and dried at room temperature.The samples were washed according to standard washing process GOST 9733.4–83. The washing solution consisted of 0.5 g of soap and 0.2 g of soda for every 100 mL of water. The fabrics in the washing solution were stirred using a magnetic stirrer at a temperature of 60 °C for 30 min.For sheet resistance measurement, silver paste contacts were applied. The distance between the contacts was 10 mm for tensile testing and 15 mm for bending testing. After application, the contacts were dried for 24 h at room temperature.

The microstructure and surface morphology of the obtained samples were studied using a Nikon Eclipse LV100 (Nikon Corporation, Shinagawa-ku, Tokyo, Japan) optical microscope, a JEOL JSM-7800F (JEOL Ltd., Akishima, Tokyo, Japan) scanning electron microscope (SEM). Raman spectra were obtained using an NT-MDT INTEGRA Spectra (NT-MDT, Moscow, Russia) Raman spectroscopy unit. Electrical properties were tested on an ASEC-03 unit (Kotel’nikov institute of radio engineering and electronics of RAS, Moscow, Russia).

## 3. Results and Discussion

In this study, the surfaces of nylon and cotton fabrics coated with rGO were investigated using a JEOL JSM-7800F scanning electron microscope. The results are shown in Figure 2 and Figure 3. Nylon has a weave of fibers with a diameter of ~19–20 microns (Figure 2a). SEM images demonstrate the presence of an adsorbed GO layer on nylon and cotton fibers (Figure 2b and Figure 4b, respectively). In the images of nylon coated with BSA/rGO, it can be seen that the nylon fibers stick together (Figure 2d). After 10 wash cycles, the nylon fibers are smoother due to the removal of unattached GO flakes (Figure 2c and Figure 3e).

Cotton fibers are composed of the interlacing of flat 10–11 µm wide threads (Figure 3a). The rGO layer on the cotton fibers can be seen in Figure 3b,d, where an intermediate BSA layer is also present. After 10 washing cycles, partial peeling of the surface layer of GO is also observed (Figure 3c,f), which correlates with the literature data [9]. It can be noted that the rGO layer on the surface of both nylon and cotton can withstand at least 10 wash cycles without damage to the coating. Thus, it can be concluded that there is good adhesion of GO to the studied textiles both in the presence of BSA layer and without it.

Figure 4 shows the Raman spectra of nylon and cotton coated with rGO. The spectra of pure nylon and cotton show different peaks related to the structure of the materials. After the deposition of GO and its reduction, the D and G bands, attributed to graphene, appear in the spectra, which is similar to the data reported in the literature [30,31]. The ratio of the intensities of the Raman peaks of these two bands (ID/IG) of GO in the case of its deposition on nylon fabric and after the removal of oxygen groups during the chemical reduction of GO remains practically unchanged. After 10 wash cycles of nylon with rGO, the intensity of the D peak exceeds the intensity of the G peak, which indicates the introduction of defects in the structure of the rGO coating. It is known that peak D is associated with the disordering of the structure and the content of defects in GO [32]. An increase in the intensity of the peak after repeated washing cycles reflects mechanical damage to the coating associated with washing. When a cotton fabric is coated with GO immediately after the reduction process, ID/IG > 1, i.e., structure disordering also occurs in the rGO film. After 10 wash cycles, this ratio remains the same, no further structural damage occurs. From this, it can be concluded that the obtained samples on both types of textiles tolerate washing well and the coatings from rGO retains its properties.

The samples were coated with a GO layer by dipping. When the deposited rGO layer was treated with hydrazine vapor, samples with the sheet resistance within the range of 1–9 kΩ/sq were obtained. Samples show good electrical conductivity, especially considering that the samples were dipped only once, in contrast to other studies [12,33]. The minimum sheet resistance for the nylon samples was 350 Ω/sq and ~1 kΩ/sq for cotton, which could be related to the weaving structure of the fabric samples.

The dependence of the electrical characteristics on the stretching of textile samples is shown in Figure 5b. It was found that the change in the resistance of nylon strips with a rGO layer depends on the GO suspension concentration. It is known that a higher concentration of GO suspension leads to an increase in absorption of GO by the fabric and an increase in coating thickness [34]. At a suspension concentration of 1 mg/mL the elongation of the sample leads to the increase in resistance (Figure 5a). When the elongation is removed, the resistance of the nylon sample does not return to its original state. This is connected with the formation of cracks in the rGO coating. Gauge factor was calculated according to the known mathematical Equation:G_f_ = (∆R/R_0_)/(∆L/L_0_)(1)

G_f_ is the gauge factor; ∆R is the resistance change; R_0_ is the resistance under zero strain; ∆L is the absolute change in length; L_0_ is original length of the sample.

Two strain range regions with different gauge factors can be identified in Figure 5a, as in study [35]: region with elongation range from 0 to 7 mm with G_f_ = 0.9; the region with elongation from 7 to 10 with G_f_ = 6.7. Significantly weaker changes in resistance were observed in the case of a thicker coating (3 mg/mL). The resistance is almost restored to its original value when elongation is removed (Figure 5a,b). It can be concluded that the increase in the resistance due to the formation of cracks is not as significant as at the lower concentration of GO (1 mg/mL). The change in resistance in this case has a complex character with a gauge factor ranging from −1.9 to 3.3.

Also, mechanical tests of cotton samples with coatings of rGO and BSA/rGO were carried out. The results are shown in Figure 5c,d. Stretching cotton samples over 6–7 mm with an initial contact distance of 10 mm leads to the rupture of fabric fibers. The increase in resistance, in this case, is associated with damage to the rGO coating. There was also damage to the connections between individual fibers, as seen in Figure 2 and Figure 3. The resistance of the sample increases irreversibly, which indicates the significant contribution of the rupture of individual fibers. The obtained dependences of the resistance on elongation with a gauge factor of 6 are similar in nature to the data given in other studies [30].

Mechanical bending tests were carried out on cotton samples with coatings based on rGO to a radius of 3–4 mm, the results of which are shown in Figure 6.

All samples were subjected to 50 bending cycles and the resistance was measured before, during, and after bending. It is evident that the resistance increases during the bending up to 10–20% and returns to its initial value after strain removal. This behavior is typical for all measured samples. The increase in resistance can be explained by the disruption of the bonds between the individual fibers during bending. The inset to Figure 5b shows an SEM image of the rGO coated nylon fabric surface after a tensile deformation cycle. The damage and formation of cracks in rGO coating caused by mechanical stretching can be observed. Further cycling does not inflict significant changes in rGO film structure. Thus, after initial bending cycles, change in resistance is stabilized around the same values. It should be noted that there is a significant increase in resistance when using a layer of BF-6 glue as an additional protective coating. On the other hand, in the case of BSA/rGO/BF-6 coating, minimal changes in the resistance were observed compared to other samples. In the case of the stretching of cotton coated with BSA/rGO/BF-6 or rGO/BF-6, an increase in resistance for 2–3 times is also observed without return to its original state.

Figure 7 shows the results of measuring the resistance of nylon and cotton samples, coated with rGO, with an intermediate layer of BSA and a protective layer of glue BF-6 (a total of 4 types of samples) during washing. In the case of nylon (Figure 7a), for all 4 samples, the resistance of the samples does not change significantly up to the 6th wash cycle. Samples without the use of an intermediate BSA layer, both with and without a layer of BF-6 glue, showed a significant increase in resistance after 7–8 washing cycles. A relatively consistent change in resistance up to 7–8 wash cycles can be associated with a change in the resistance of the rGO film caused by a decrease in its thickness. A sharp increase in resistance after 7–8 cycles is associated with a disruption of the electrically conductive network of rGO. For samples with adhesive BSA layer, a slight increase in resistance is observed after the completion of the 40th wash cycle. The sample with BSA/rGO coating showed the best sustainability to washing. This may indicate that the presence of the BSA layer significantly enhances the adhesion of GO particles to the nylon surface.

In the case of cotton, as shown in Figure 7b, for samples with a BF-6 protective coating, the resistance does not change significantly even after 50 washing cycles. In samples of cotton fabrics with rGO and BSA/rGO, a significant increase in resistance occurs during washing. Thus, the BSA layer in the case of cotton fibers does not improve the adhesion of GO flakes. Overall, the data on the stability of pristine GO on nylon and cotton correlates with data obtained by other authors [9,20]. It can be concluded that the use of BF-6 as a protective layer can significantly improve the mechanical stability of e-textiles based on cotton. On the contrary, use of BSA significantly improves the stability of nylon e-textiles, but does not provide significant benefits for cotton.

To demonstrate the conductivity of the electronic textiles, samples were used to connect the LEDs to a power source (Figure 8) in both the original and bended states.

The resulting conductive samples of nylon and cotton were tested for sensitivity to relative humidity (Figure 9), temperature (Figure 10), and tensile stress associated with the human heart rate (Figure 11). As shown in Figure 10, there is a strong dependence on resistance with the temperatures ranging from −80 to +60 °C. This indicates that the temperature sensor can be used for wearable monitoring devices in Arctic conditions. The temperature sensitivity was approximately the same for all types of coatings and materials, and correlates with the data in the literature [12]. Humidity sensors based on obtained samples showed good sensitivity of resistance towards humidity change. The strongest changes in resistance were observed for nylon samples.

The heart rate calculated from Figure 11a,b for a sitting and walking person is 66–78 and 84–90 beats per minute, respectively. Results coincide with the readings of a simultaneously operated commercial heart rate sensor (72 and 90 beats per minute, respectively).

## 4. Conclusions

Samples of the conductive textiles, nylon and cotton, with a layer of rGO were investigated. It was shown that with the use of the intermediate layer of BSA to nylon samples there was an improvement to the adhesion of the rGO. Depending on the thickness of the deposited rGO layer, it was possible to obtain samples with minimum resistance values of 350 Ω/sq for nylon, and about 1 kΩ/sq for cotton. Mechanical stretching of samples led to an increase of the resistance for 2–3 times with a change in length of 6–9 mm using an initial sample size of 10 mm. An increase in the concentration of GO flakes in the suspension from 1 mg/mL to 3 mg/mL led to a decrease in the resistance change on tensile stress from 3.5 to 1.7 times, due to the increase of coating thickness. Cyclic (50 cycles) mechanical bending tests up to a radius of 3–4 mm of samples with different coatings showed that the resistance slightly (up to 10–20%) increases during bending and steadily returns to its original state. For the first time, it was demonstrated that the adhesive layer of BF-6 (rGO/BF-6 and BSA/rGO/BF-6) can be used as an additional protective coating. This layer increased the electrical resistance of the samples, and, at the same time, made it possible to reduce changes in resistance during bending and stretching. Washing of the conductive textiles has shown their stability. Presence of the BSA layer enhances the adhesion of GO particles to the nylon surface. In contrast, BSA layer in the case of cotton fibers does not significantly improve the adhesion of GO flakes. Protective layer of BF-6 improves the mechanical stability of e-textiles based on cotton, but not has significant effect on nylon textiles. It has been shown that resistance of textile samples coated with rGO are sensitive to changes in temperature, humidity, and mechanical stress associated with the human heart rate. The obtained e-textiles can be used for wearable electronics, including smart clothing for monitoring vital signs in the Arctic.

## Figures and Tables

**Figure 1 materials-14-01999-f001:**
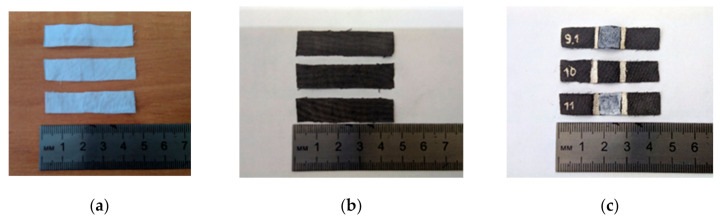
(**a**) Pristine cotton samples with a size of 1 cm × 4.5 cm. (**b**) Cotton samples after coating with bovine serum albumin (BSA) and reduced graphene oxide (rGO). (**c**) Cotton/BSA/rGO and cotton/BSA/rGO/BF-6 samples with silver paste electrodes after 10 washing cycles.

**Figure 2 materials-14-01999-f002:**
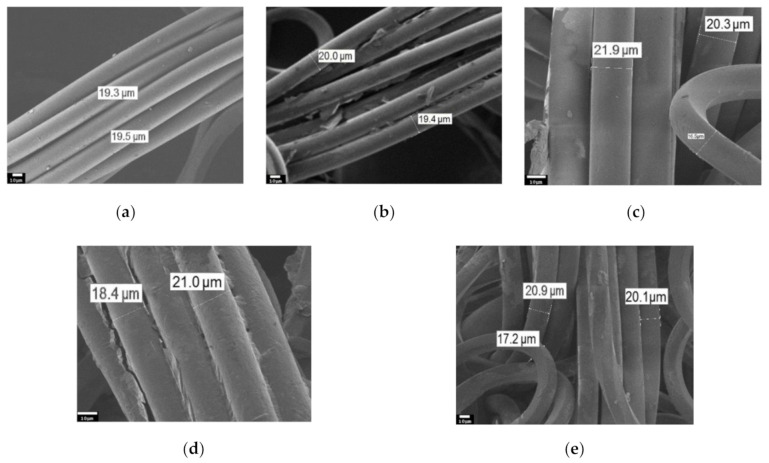
Scanning electron microscopy (SEM) images of nylon fibers (500× magnification): (**a**) Pristine material; (**b**) Nylon/rGO; (**c**) Nylon/rGO after 10 washing cycles; (**d**) Nylon/BSA/rGO; (**e**) Nylon/BSA/rGO after 10 washing cycles.

**Figure 3 materials-14-01999-f003:**
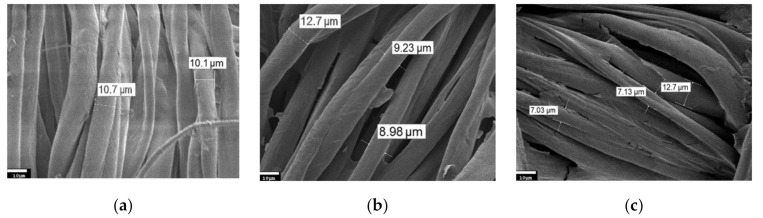
SEM images of cotton fibers (500× magnification): (**a**) Pristine material; (**b**) Cotton/rGO; (**c**) Cotton/rGO after 10 washing cycles; (**d**) Cotton/BSA/rGO; (**e**) Cotton/BSA/rGO after 10 washing cycles.

**Figure 4 materials-14-01999-f004:**
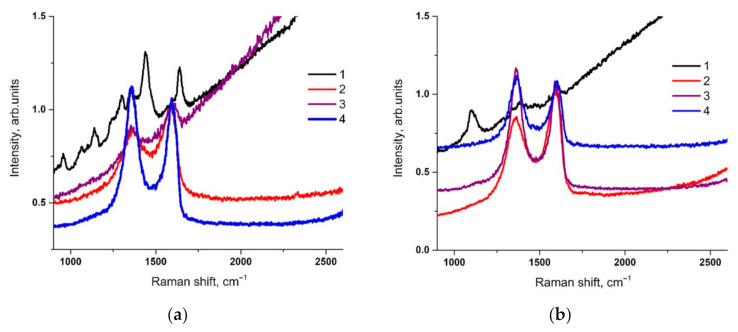
Raman spectra for samples: (**a**) Nylon, (**b**) Cotton: 1–pristine material; 2–one coated with graphene oxide (GO); 3–after reduction of GO; 4–after 10 washing cycles.

**Figure 5 materials-14-01999-f005:**
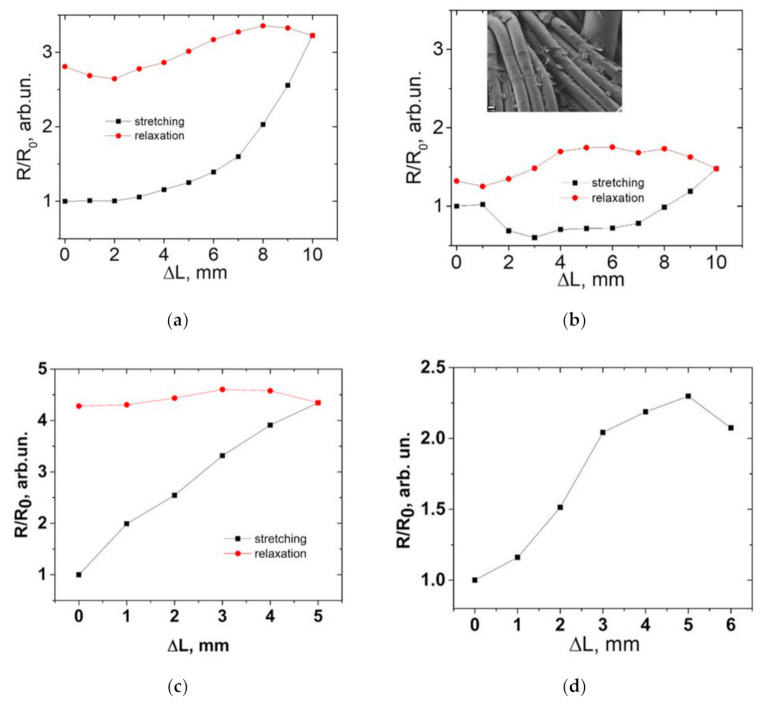
The normalized resistance for nylon/rGO samples with a GO concentration in the suspension (**a**) 1 mg/mL and (**b**) 3 mg/mL, depending on the stretching and return to the original state. Inset is an SEM image nylon/rGO surface after a stretching cycle. (**c**,**d**) The normalized resistance of cotton samples coated with (**c**) rGO and (**d**) BSA/rGO under stretching.

**Figure 6 materials-14-01999-f006:**
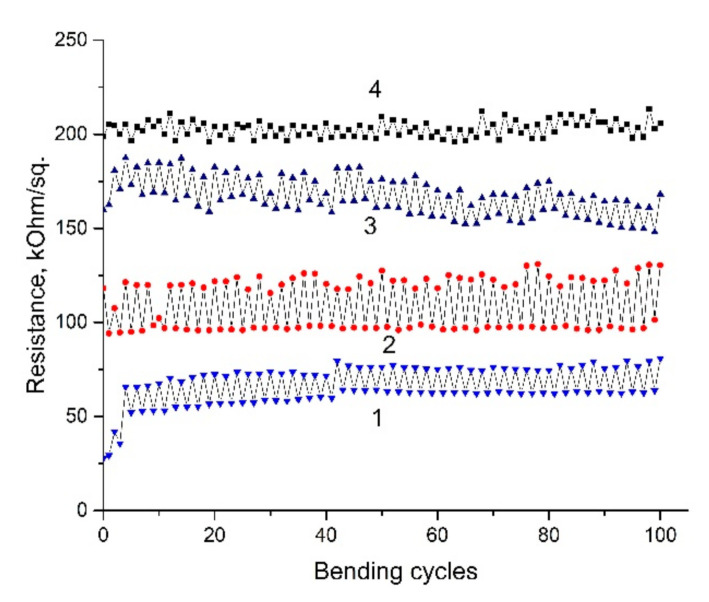
Resistance of cotton samples with different coatings before, during and after bending (50 cycles): 1-rGO, 2-BSA/rGO, 3-rGO/BF-6, 4-BSA/rGO/BF-6. The bending radius was 3–4 mm.

**Figure 7 materials-14-01999-f007:**
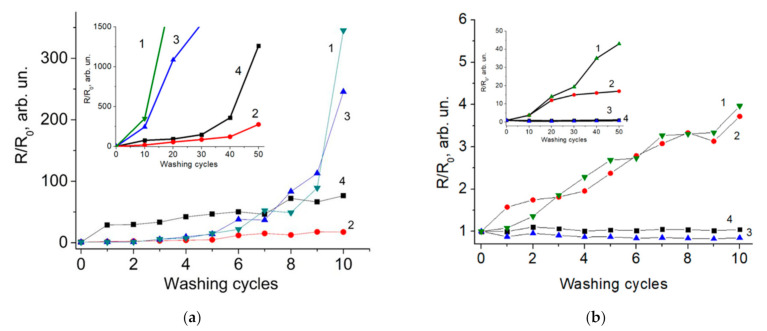
Resistance of samples after 10 washing cycles: (**a**) Nylon; (**b**) Cotton. 1-rGO, 2-BSA/rGO, 3-rGO/BF-6, 4-BSA/rGO/BF-6. Insets inside the images reflects the change in resistance for a further 50 washes, respectively for nylon and cotton samples.

**Figure 8 materials-14-01999-f008:**
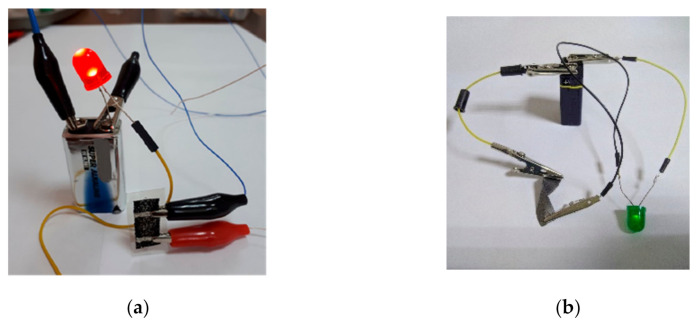
Demonstration of the textiles with rGO conductivity via connecting to LEDs: (**a**) nylon; (**b**) cotton samples.

**Figure 9 materials-14-01999-f009:**
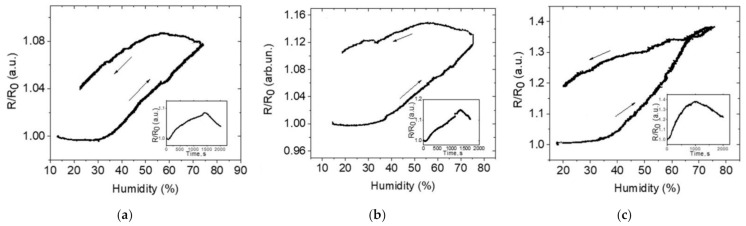
The resistance depending on relative humidity for samples: (**a**) cotton/rGO, (**b**) cotton/BSA/rGO, (**c**) nylon/rGO.

**Figure 10 materials-14-01999-f010:**
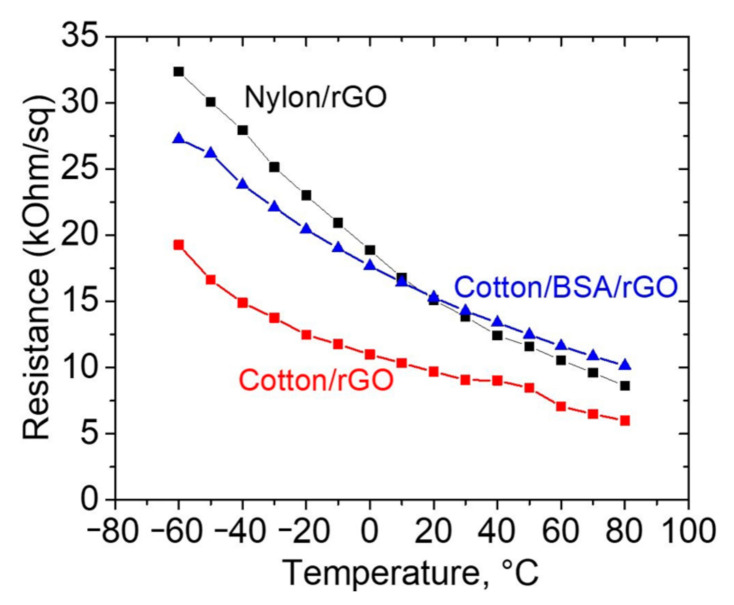
The resistance of cotton and nylon samples with rGO and BSA/rGO depending on temperature. The cotton/rGO resistance values multiplied by 3 for better visualization.

**Figure 11 materials-14-01999-f011:**
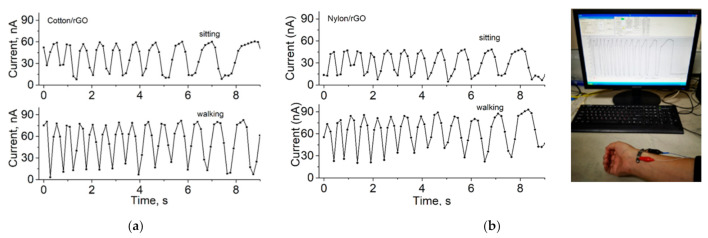
Heart rate measuring current-time characteristics for a sitting and walking person: (**a**) cotton/rGO; (**b**) nylon/rGO.

## Data Availability

The data presented in this study are available on request from the corresponding author. The data are not publicly available due to data protection policy of the university.

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
