# Peer review of "Electrical Properties of Textiles Treated with Graphene Oxide Suspension"

_materials, 2021, doi:10.3390/ma14081999_

Round 1

Reviewer 1 Report

Interesting results are reported and the manuscript was reviewed for publication in Materials-MDPI Journal. However, to warrant publication, the authors need to consider following major points -

  1. Introduction is too short and lacks many aspects to be discussed such discussion on nylon and cotton-based electronic textile, the textile coatings based on GO and rGO and so on need to be included. A bridge between the existing literature and work reported in literature is missing. In last paragraph, please describe the novelty of the work such as how this work is important, why is advanced from the existing literature and so on…
  2. In introduction, authors need to discuss also about other electrical flexible substrates such as PDMS in addition to textile used for strain sensing. There are several papers on them. Few citable papers are – [https://doi.org/10.1002/vnl.21799], [https://doi.org/10.1016/j.polymer.2020.122221],[https://doi.org/10.1016/j.apsusc.2017.04.068], [https://doi.org/10.1016/j.carbpol.2013.03.052]
  3. The discussion in result and discussion section throughout the paper is poor. The validation of results obtained from the work need to be discussed with those reported in literature is missing. So, more papers are required to be included and discussed in the concerned section.
  4. Figure-1 is well known and can be deleted or shifted to the supporting information section.
  5. GO was prepared from modified Hammers method reported in literature. So, what is the novelty in synthesizing GO in this work. Similarly, what is novelty in preparing rGO?
  6. To estimate the durability of the specimen, 10 cycles wash cycles were obtained. I think this is too few cycles. It should be atleast 50 wash cycles to observe the effect of washing on performance of e-textile studied in this work.
  7. In Raman spectra studied, authors reports in #line 143-144 “After 10 wash cycles of nylon with rGO, the intensity of the D peak exceeds the intensity of the G peak”. Please describe this result?
  8. It will be interesting to estimate Gauge factor of e-textile printed with GO and rGO from the measurements? Moreover, the fracture strain study of e-textile will be appreciated.
  9. During cycling tests, the resistance increases due to formation of cracks, please use SEM to study development of cracks and types of cracks on the electrode based on GO and rGO?
  10. In Figure 8a, the relative resistance increases significantly especially after 8th washing cycle. Please explain why?

Author Response

Dear, reviewer!

Thank you for a very detailed review of our article, as well as for pointing out the shortcomings in our study. It is a great opportunity for us to receive feedback from the world scientific community since we are geographically located in a remote area.

Best regards, autors.

Reviewer 2 Report

The manuscript describes the properties of nylon and cotton-based electronic textiles coated with reduced graphene oxide (rGO). The effect of bovine serum albumin (BSA) as an adhesive layer and BF-6 glue as a protective layer on resistance to mechanical stress and washing has been investigated. The content is comprehensive and interesting.

This manuscript can be accepted after major revision. Thank you for the opportunity to review your manuscript.

Some comments are listed below:

  1. Please add some numerical data (best results) to abstract to increase the attractiveness of this article.
  2. Please correct Figure 1a to better quality.
  3. Please seek guidance from native English speaker if possible.
  4. Hammers method (line 60 in Introduction) change to Hummers method.
  5. Figure 3 and Figure 4 (Is it possible to show these figures with different magnification or write in comments why you use only one 500x magnification. Can you changed a,b,c,d etc. in all figures to be better visible.
  6. Can you add for example some HRTEM figures ?
  7. Figure 5 (Raman spectra) can you prepare line 1 (5a and 5b) and line 3 (5a) with better quality ?
  8. Line 233 please cut ","

Author Response

Dear, reviewer!

Thank you for your review. Working on the problems that you identified allowed us to improve the article. It is a great opportunity for us to receive feedback from the world scientific community since we are geographically located in a remote area.

Best regards, autors.

Reviewer 3 Report

This manuscript presents the modification of textiles (nylon and cotton) with graphene oxide suspensions in order to obtain electronic textiles for sensors (monitoring of vital signs embedded in clothing). In fact, the paper is interesting, presents some novelty, it is well written and the results are well discussed, but it lacks a Materials and Methods section (authors must check the Instructions for Authors of the journal, I believe that this section is mandatory). Besides, some flaws can be pointed out: 
1. the authors mention the "Hammers method". I believe that the correct is "Hummer's method". 
2. The authors do not introduce the abbreviation of graphene oxide (GO), only reduced graphene oxide (rGO). 
3. The last paragraph of the Introduction must be revised.
4. At the results section, several units appear in a wrong manner such as "kΩ/□". 
5. Page 5, line 157: "It was found that the change in the resistance of nylon strips with an rGO layer depends on the GO suspension concentration, i.e. on the thickness of the nylon coating with GO flakes." ??? 
6. The authors did not state how they verified the GO concentration. It was via UV-VIS? HPLC?
7. 1 and 3 mg/mL seems to be very high concentration values for GO suspensions. There are published papers which deposited graphene oxide (not reduced) onto polymeric substrates with concentrations in a range of ca. 25 mg/L. Why the authors selected these concentrations?

In my opinion, before consideration for publication, the paper must be revised in a overall manner, to assure that all information is delivered properly to the readers. This includes to write a materials and methods section with enough details in order to allow the study performed by the authors to be reproduced by other researchers. 

Author Response

Dear, reviewer!

Thank you for your review. We hope that the work on the shortcomings that you noted allowed us to improve our article. It is a great opportunity for us to receive feedback from the world scientific community since we are geographically located in a remote area.

Best regards, autors.

Round 2

Reviewer 1 Report

I have no further comments. 

Author Response

Dear, Reviewer!
We hope that thanks to your notes on the shortcomings of our article, we have significantly improved its quality. Thank you for your review.

Best regards, authors

Reviewer 2 Report

Now manuscript is ok. I accept it.

Author Response

Dear, reviewer!
We hope that thanks to your notes on the shortcomings of our article, we have significantly improved its quality. Thank you for your review.

Best regards, authors

Reviewer 3 Report

The authors addressed all my suggestions and therefore I believe that the paper can be accepted for publication. However, some minor issues must be addressed such as some lack of uniformity on the information such as: authors must define every acronym (e.g. CNT = Carbon nanotubes (CNT)).

Author Response

Dear Reviewer!

Thanks for your comment, we have double-checked all the acronyms. The acronym for CNT is entered on line 31:
"In particular, there are many research articles on the use of carbon nanotubes (CNTs) [5] ..."

We hope that thanks to your notes on the shortcomings of our article, we have significantly improved its quality. Thank you for your review.

Best regards, authors